# The Association between Telomere Length and Head and Neck Cancer Risk: A Systematic Review and Meta-Analysis

**DOI:** 10.3390/ijms25169000

**Published:** 2024-08-19

**Authors:** Dimitrios Andreikos, Efthymios Kyrodimos, Athanassios Kotsinas, Aristeidis Chrysovergis, Georgios X. Papacharalampous

**Affiliations:** 1School of Medicine, Democritus University of Thrace, 68100 Alexandroupoli, Greece; 2First Department of Otorhinolaryngology—Head and Neck Surgery, Faculty of Medicine, National and Kapodistrian University of Athens, 114 Vas Sofias Avenue, 11527 Athens, Greece; ekirodim@med.uoa.gr; 3Laboratory Histology–Embryology, Faculty of Medicine, National and Kapodistrian University of Athens, 75 Mikras Asias Street, Goudi, 11527 Athens, Greece; 4ENT Department, Athens General Hospital “ELPIS”, 7 Dimitsanas Street, 11522 Athens, Greece; achrysovergis@gmail.com (A.C.); poulador@yahoo.gr (G.X.P.)

**Keywords:** telomere length, telomere, association, cancer, head and neck cancer, meta-analysis

## Abstract

Telomeres play a crucial role in maintaining chromosomal integrity and regulating the number of cell divisions and have been associated with cellular aging. Telomere length (TL) has been widely studied in manifold cancer types; however, the results have been inconsistent. This systematic review and meta-analysis aims to analyze the evidence on the association between TL and head and neck cancer (HNC) risk. We comprehensively searched the literature in PubMed, Cochrane Library, and Scopus and identified nine eligible studies, which yielded 11 datasets. The odds ratios (ORs) and 95% confidence intervals (CIs) were used to ascertain the strength of the association. On the basis of the median TL, we defined two groups, short TL and long TL, with the latter being the reference group. Our analysis found a significant relationship between short TL and increased HNC risk (OR 1.38, 95% CI: 1.10–1.73, *p* = 0.005), while significant heterogeneity among the studies was noted. The subgroup analysis on HNC subtypes revealed a significant association between short TL and oral cancers (OR 2.08, 95% CI: 1.23–3.53, *p* = 0.007). Additionally, subgroup analysis indicates that adjustments for age, sex, and smoking did not affect the significance of our findings. In conclusion, our meta-analysis found evidence for an association between short TL and HNC risk, which could indicate that TL might act as a potential biomarker for HNC risk, but high-quality prospective studies are imperative to validate our findings.

## 1. Introduction

Head and neck cancers (HNCs) constitute the 7th most common cancers worldwide, with up to 900,000 reported new cases and approximately 400,000–600,000 deaths annually. Their incidence has been increasing and also has been projected to maintain an upward trend [1]. The types of cancer included in HNCs are malignancies of the larynx, hypopharynx, pharynx, oral cavity, salivary glands, and nasal cavity, while the most common histological subtype is squamous cell carcinoma [1,2]. Even though there have been significant developments in therapeutic modalities and an increase in survival in the last decades, the overall prognosis remains poor [3]. The 5-year survival rate is highly dependent on the stage of the malignancy at the time of initial diagnosis, with an excellent prognosis of up to 90% in many HNCs at stage I [4]. Unfortunately, the majority of HNCs are diagnosed at advanced stages with often abysmal prognosis [2]. Therefore, a biomarker that may be predictive of the risk of HNC could potentially help in lowering the stage of the malignancy by the time of diagnosis, thus improving survival rates.

Telomeres are located at the terminal ends of chromosomes and are complex nucleoprotein structures with a tandem repeated hexameric DNA sequence (5′-TTAGG-3′) [5,6]. The telomeres that cap the chromosomes are approximately 10–15 k base pairs and help maintain the structural integrity of linear DNA by preventing chromosomal degradation, as they safeguard against end-to-end fusions and accidental recombination [7]. Telomeres shorten by 30–200 bp during each cycle of DNA replication and when a threshold is reached, pathways of apoptosis or cellular senescence are activated [7].

Among the commonly used techniques for measuring TL are quantitative polymerase chain reaction (qPCR), terminal restriction fragment (TRF) analysis, and flow cytometry with fluorescent in situ hybridization (Flow-FISH) [8,9,10]. Physiologically, TL is affected by age, sex, genetic factors, parental factors, and lifestyle factors [11,12]. According to studies, individuals with higher levels of physical activity and healthier diets tend to have longer telomeres [13]. Stress and lifestyle factors including smoking and excessive alcohol consumption have been associated with shorter TL [14,15]. The length of the telomeres has been documented to decrease with normal aging, but many studies have also suggested an independent association of telomere length (TL) with a diverse set of pathologies including pulmonary and cardiovascular diseases and cancer [11,16,17,18,19,20].

The association between telomere length and cancer is considered to be one of the most interesting issues in the current literature, with promising clinical applications. Basic research indicates that cancer is associated with shorter telomeres. A proposed mechanism involves the reactivation of telomerase leading to maintenance of TL above the apoptotic threshold, while genomic instability due to the shorter telomeres induces carcinogenic mutations [21,22,23,24]. After examining the published meta-analyses, a significant relationship between cancer risk and short TL was demonstrated in urological cancers, lung cancer, and cancers of the digestive system [25,26,27]. However, in meta-analyses of breast and colorectal cancers, the association did not reach statistical significance [28,29], and in melanoma, shorter TL was associated with decreased risk [30]. In meta-analyses that included various types of cancers, despite the significant associations discovered in certain cancers, the overall relationship was found to be non-significant [26,31]. This could be indicative of diversity in telomere biology between different types of cancer. Therefore, in conjunction with the evidence from basic research, the validity of TL as a potential cancer biomarker and its interpretation should be ascertained in a clinical setting individually in each cancer type.

In HNC studies, telomere biology has been suggested to play a significant role, as HNCs have been associated with pathogenic mutations in the protein-coding genes related to telomerase activation, telomerase activity, and TL [32,33,34,35]. Furthermore, there are manifold studies quantitatively examining the association of TL and HNC risk in humans, but the results are inconsistent and have yet to be synthesized. A meta-analysis could ascertain the significance of the association, potentially indicating a role for TL as a risk biomarker for HNCs. In the present study, we conducted the first systematic review and meta-analysis of the published studies on the association between TL and HNC risk.

## 2. Materials and Methods

### 2.1. Literature Search Strategy

The meta-analysis was conducted following the PRISMA standards and is in accordance with the PRISMA 2020 statement [36].

We planned and performed a literature review using the electronic databases PubMed, Cochrane Library, and Scopus to identify the relevant papers published until 28 February 2024. The search of the literature was conducted using the terms “Telomere”, “Telomere length”, “Head and neck cancer”, the MeSH term for malignancy of the head and neck region, “Head and Neck Neoplasms”, and logical operators to combine them.

All papers were initially evaluated based on the title and abstract. We thoroughly examined the full text of any paper deemed eligible and reviewed the bibliographical references for additional potentially relevant papers. The eligibility of studies was assessed by two independent reviewers and any conflicts were resolved by consensus.

#### 2.1.1. Study Inclusion/Exclusion Criteria

The inclusion criteria were the following: (1) Studies must include patients with head and neck malignancies and controls (healthy patients or healthy tissues). (2) The study must provide the odds ratio (OR) and 95% confidence intervals (95% CI) for the relationship between TL and head and neck malignancies. Alternatively, a study was considered eligible if the OR could be calculated from the data. (3) No time restrictions were used. (4) Case–control, nested case–control, and cohort studies were included.

The following criteria were used to exclude studies: (1) Papers examining malignancies of the thyroid gland, the parathyroid glands, and the nasopharynx. (2) Studies on non-malignant neoplasms or data within a study pertaining to benign or pre-malignant neoplasms. (3) Case reports, editorials, reviews, and meta-analyses. (4) Studies not published in the English language.

#### 2.1.2. Data Extraction

For all eligible studies, we extracted the following data: first author, year of publication, region, study design, sample size, sex and age distribution of the participants, distribution of cases and controls (in case–control studies), cancer type, method of TL measurement, source of the DNA, ORs and the corresponding 95% CI for short vs. long TL in relation to cancer, and the adjustments in the calculation of the ORs. If an article included multiple datasets, we considered them independent studies. The data were inputted into a standardized data sheet by two independent reviewers and a consensus approach was used to resolve any conflicts.

### 2.2. Statistical Analysis

The study subjects of each study were divided into two groups (short TL and long TL) based on TL. To investigate the relationship between TL and HNC risk, we used ORs and the corresponding 95% CIs. The median TL was considered the cut-off point for defining the short TL group when calculating the OR. The short TL group was defined as the exposure group and the long TL group as the reference group. If the OR was not given, where possible, we used the available data to construct dichotomous categories and calculate the ORs and corresponding 95% CI.

The statistical heterogeneity between studies was assessed using the X^2^-based Q test and the I^2^ statistic. Heterogeneity was considered significant if *p* < 0.10 in the Q test or if I^2^ > 50%. If heterogeneity was not significant, the “fixed effects model”, based on the Mantel–Haenszel method, which assumes homogeneity of effect size, was used to combine the studies. Otherwise, the “random effects model”, based on the DerSimonian and Laird method, which produces wider confidence intervals when the results of the individual studies differ, was employed to combine the studies. We performed a subgroup analysis in which the studies were stratified based on cancer type, region, DNA source, adjustment for age, sex and smoking status, and number of subjects. Publication bias was examined using a Funnel Plot, and a sensitivity analysis was performed.

Studies that were excluded based on the eligibility criteria but examined the relationship between TL and cancer risk are discussed in the text, although they are not included in the meta-analysis.

All analyses were conducted using RevMan, Review Manager 5 (RevMan 5); Date of publication, 2020; Edition, 5.4. A two-tailed *p*-value of less than 0.05 was defined as the criterion for statistical significance.

### 2.3. Study Selection

The study selection process adheres to the Preferred Reporting Items for Systematic Reviews and Meta-Analyses (PRISMA) and is presented in detail in the flow chart in Figure 1.

The initial search in PubMed, Cochrane Library, and Scopus yielded 589 studies, which were subsequently refined to 525 after the removal of the duplicates. The studies’ titles and abstracts were examined, and 499 records were excluded as irrelevant to the topic, leaving 26 articles to be retrieved.

Subsequently, the full texts of the 26 articles were successfully retrieved. These articles were meticulously assessed for eligibility. Upon evaluation, three articles were excluded for not providing ORs or relevant data, two for lacking TL data distribution, five for not including a healthy control group, four studies for being conducted on cell lines, and three were excluded due to other criteria.

Finally, nine studies, which included 11 datasets, met the eligibility criteria and were included in our systematic review and meta-analysis.

## 3. Results

### 3.1. Study Characteristics

The key characteristics of the included studies are summarized in Table 1 and Table 2.

In terms of geographical distribution, five studies were conducted in the United States, two in China, one in Japan, one in Brazil, one in Italy, and one in Lithuania. The mean age of the participants ranged from 55 to 66 years of age. One study did not report on the mean age of its subjects, and in one study, which divided the control group into young and old subgroups, the latter was used as the control group after consideration of the mean age of the control groups in the other studies. Gender distribution varied among the studies, from 59% to 96% male participants in the case group. However, the most prevalent distribution was approximately 74% males in the case group.

Among the selected studies, six investigated a specific HNC subtype, three reported data on HNC as a collective entity, and one reported on HNCs excluding oral cancers. Eight studies sampled DNA from peripheral leukocytes and two studies extracted DNA from tumor tissue samples. TL measurements were conducted using quantitative polymerase chain reaction (qPCR) in eight studies, while Southern blot was used to measure DNA in two studies. The median TL was considered as the cut-off point for defining the short TL group when calculating the OR. In one study, in which the OR was given based on the shortest quartile of TL [43], utilizing the study’s data and considering a normal distribution of TL, the OR and 95% CIs based on the median were derived. The ORs and corresponding 95% CIs were not provided by the study but were calculated using provided data from the studies in two cases [38,44]. All the included studies followed a retrospective study design.

### 3.2. Results of Studies

#### 3.2.1. Meta-Analysis

Our results suggest a significant relationship between HNC risk and TL (*p*-value = 0.005). There was significant heterogeneity between the studies with I^2^ statistic = 81% and chi^2^ = 52.99. The summary OR from 11 datasets found in nine studies was calculated at 1.38 (95% CI: 1.10–1.73) (Figure 2).

#### 3.2.2. Subgroup Analysis

To further explore the relationship between TL and HNC, we performed a subgroup analysis using selected categories. We analyzed four studies with patients who exclusively had cancer of the oral cavity (Figure 3). In those particular studies, there was a significant association (*p*-value = 0.007) between short TL and OCC risk (OR = 2.08, CI 95% [1.23–3.53]). On the contrary, in the subgroup of studies with non-exclusively oral cavity cancer patients, the association between short TL and cancer risk (OR = 1.17, CI 95% [0.89–1.55]), was not statistically significant (*p*-value = 0.27).

In studies in which the DNA source was PBL, a significant association (*p*-value = 0.01) between short TL and HNC risk was demonstrated (OR = 1.33 CI 95% [1.06–1.67]) (Figure 4). A non-significant association was found in the studies where tissue samples were used as the DNA source (OR = 2.56 CI 95% [0.47–13.87], *p*-value = 0.28).

We also conducted a subgroup analysis on the basis of the continent of origin: we found a significant association in studies from America (OR 1.69 CI 95% [1.07–2.66], *p*-value = 0.02), while no significant association was found in Europe (OR 0.85 CI 95% [0.54–1.32], *p*-value = 0.53) and in Asia (OR 1.22 CI 95% [0.87–1.70], *p*-value = 0.25) (Figure 5).

Additionally, we analyzed the impact of adjustment for sex and age on the association of TL and HNC risk. In studies with OR adjusted for sex and age, a significant relationship was demonstrated (OR = 1.32 CI 95% [1.06–1.65], *p*-value = 0.01) (Figure 6). The relationship was also significant in the unadjusted study (OR = 6.49 CI 95% [1.58–26.61], *p*-value = 0.009). We also performed a subgroup analysis with studies adjusted for sex, age, and smoking status; adjusted only for sex and age; and unadjusted. This subgroup analysis found a significant relationship in the first subgroup (OR = 1.30 CI 95% [1.04–1.64], *p*-value = 0.02) and in the third subgroup (OR = 6.49 CI 95% [1.58–26.61], *p*-value = 0.009), while the relationship in the second subgroup was non-significant (Figure 6).

We examined the impact of sample size on the relationship between short TL and HNC risk (Figure 7). We found no significant association in studies with a sample size < 100 (OR = 2.56 CI 95% [0.47–13.87], *p*-value = 0.28), while the association was significant in studies with a sample size > 100 (OR = 1.33 CI 95% [1.06–1.67], *p*-value = 0.01).

### 3.3. Publication Bias and Sensitivity Analysis

To assess the meta-analysis for publication bias, we created a Funnel Plot (Figure 8). The Funnel Plot, following visual inspection, indicates no publication bias among the larger studies. Among the smaller studies, the Funnel Plot shows some asymmetry, which could be evidence of publication bias. We performed a sensitivity analysis by calculating the summary OR while successively removing each study. We found that the significance of the overall result did not change by the removal of any one study (Appendix A). The analysis was performed by two independent authors and any conflicts were resolved by consensus.

## 4. Discussion

The current literature regarding the relationship between cancer risk and short TL presents heterogeneous and controversial results. Therefore, the delineation of the association in each type might be crucial for the development of clinical applications, and our study is the first meta-analysis to investigate the association between TL and HNC.

The published literature on the association between HNC and TL was thoroughly reviewed and meta-analyzed. The results suggest a statistically significant association between HNCs and TL (OR =1.38,CI 95% [1.10–1.73]). Specifically, patients with HNC had significantly shorter TL compared to controls.

In an attempt to overcome the aforementioned heterogeneity observed in the literature, we conducted a subgroup analysis on the basis of HNC subtypes. In studies with patients who had exclusively oral cancers, a significant positive association of risk with short TL was demonstrated (OR = 2.08, CI 95% [1.23–3.53], *p*-value = 0.007). On the contrary, in the subgroup of studies with non-exclusively oral HNC patients, the positive association with short TL did not reach statistical significance (*p*-value = 0.27). Unfortunately, not enough data were available to conduct a subgroup analysis of other HNC subtypes. Interestingly, while there were no data on the distribution of TL in each cancer subtype in the non-exclusively oral cancer studies, a non-significant negative association of risk and short TL was found in the studies with no oral cancers included. One possible interpretation of this finding is that oral cancers might be responsible for the positive association in the non-exclusively oral HNC studies, but further studies are imperative to reach a definitive conclusion.

In our meta-analysis, most studies measured the TL in peripheral blood lymphocytes, and not in cancer cells themselves. Peripheral leukocyte TL has been associated with TL in other cell types [46,47], and its relationship with cancer has been widely studied in a clinical setting [25,26,28,29,30]. However, telomere biology in cancer cells has been shown to follow complicated dynamics and diverse results have been shown depending on the type of cancer [31,48]. We performed a subgroup analysis to compare studies that measured TL in leukocytes and in cancer cells. Additionally, it is critical to examine the relationship using only peripheral leukocytes as the DNA source, since it is less invasive compared to cancer tissue samples and enables the utilization of TL as an accessible biomarker. In the leukocyte subgroup, a statistically significant association between HNC and shorter TL was found, while in the intra-tumoral group, even though a similar association was indicated, it did not reach statistical significance, which could be attributed to the limited number of studies and the smaller sample sizes. The results of our subgroup analysis indicate that in HNCs, peripheral leukocyte TL is associated with HNC risk and that further studies are imperative to confirm the association in intra-tumoral TL. Furthermore, as cell-free DNA from HNCs reflects the intra-tumoral environment, research to confirm the association with intra-tumoral TL may enhance the potential of TL as a biomarker in the rapidly expanding field of liquid biopsies [48].

The results of a subgroup analysis conducted on the basis of the continent of origin found variations in the relationship between regions. The association of short TL and HNC was positive and significant in the American subgroup (OR 1.69 CI 95% [1.07–2.66], *p*-value = 0.02), positive but non-significant in the Asian subgroup (OR 1.22 CI 95% [0.87–1.70], *p*-value = 0.25), and negative and non-significant in Europe (OR 0.85 CI 95% [0.54–1.32], *p*-value = 0.47). The data were insufficient for a subgroup analysis of the basis of ethnic groups, thereby the results may be interpreted as either a consequence of cultural or genetic factors, as TL has been associated with both racial and lifestyle/cultural factors [49,50]. Another possible interpretation could be a positive association independent of region. This assumption is based on a positive association being found in all studies in the Asian and European subgroups—excluding those that did not contain any oral cancers.

We also examined the effect of adjusting for age and sex through subgroup analysis. TL has been extensively investigated and the inverse relationship between age and TL has been clearly established in the current literature [51]. In HNCs, a significant association between age and risk has been observed [52], and the median age of the HNC is high compared to the population median, often found to be between 60 and 65 years of age [53,54]. Therefore, it is a critically significant result that our subgroup analysis showed a significant relationship between HNC risk and short TL in the adjusted group (OR 1.32, *p*-value = 0.01), with an OR similar to that of the main meta-analysis (OR 1.38, *p*-value = 0.005). This finding minimizes the possibility of the effect being due to an age-related confounding factor.

Moreover, we performed a subgroup analysis on the basis of adjustment for age, sex, and smoking status; age and sex; and no adjustments. It was been widely recognized that smoking is a major risk factor associated with HNC, as it is estimated that 70–80% of new HNCs are associated with alcohol and smoking. There is evidence that suggests that smoking is independently associated with TL [55]; therefore, it may potentially act as a confounding factor [56,57]. Our subgroup analysis showed that in the studies that had been adjusted for age, sex, and smoking status, a statistically significant association between HNC and short TL was present (OR = 1.30 CI 95% [1.04–1.64], *p*-value = 0.02), indicating decreased likelihood that smoking might be a confounding factor.

As far as the effect of sample size on our results is concerned, number 100 was set as a cut-off point for sample size in the subgroup analysis. In our meta-analysis, most studies had a sample size equal to or greater than 50—above the point where sample error may cause noticeable bias [58]. In the interest of caution in examining for potential biases, we doubled the indicated cut-off point to 100 and found in the subgroup with a sample size over 100 a statistically significant positive relationship between HNC and short TL (OR = 1.33 CI 95% [1.0–1.67], *p*-value = 0.01). The same relationship was found in the main meta-analysis, thus indicating that sample size-related bias was unlikely.

The overall result of our meta-analysis indicated a statistically significant conclusion. However, our study had some limitations.

(a)Although we conducted a thorough literature search and carefully extracted the data from studies that satisfied our inclusion criteria, certain articles that investigated the relationship did not publish the necessary data for inclusion in our meta-analysis [32,59].(b)Significant heterogeneity was noted between the included studies. Although we conducted subgroup analyses in order to investigate the source of this heterogeneity, this particular problem may stem from factors that have not been accounted for in our analysis.(c)In some subgroups, the small number of studies and small sample size limited the ability to derive statistically significant results. Specifically, an important limitation was the lack of data to investigate the relationship of TL with HNC subtypes, with the exception of oral cancers.(d)While across all studies, the definitions of short and long TL were consistently based on the median telomere length, complete homogeneity was not achieved. In one study [40], the median of all subjects was used as the cutoff point and the necessary data were not provided for the derivation of the results using the median TL of controls as the cutoff, the cutoff used in the other studies.(e)The studies did not include enough data on TNM/UICC stage, p16 status, or HPV status for a meaningful analysis or subgroup meta-analysis to be performed. Future studies should include such critical data.(f)The studies included in our meta-analysis had widely heterogeneous approaches to reporting TL measurements (RTL—relative telomere length; age-adjusted relative LTL—leukocyte telomere length; normalized TCRa (NTCR); T/S median; etc.) and some did not include any direct measurements of TL. Therefore, no meaningful analysis could be conducted of the range/median/standard deviation of TL.(g)Finally, one other potential limitation could be the design of the included studies, as they generally followed a case–control pattern, which has been traditionally associated with an increased risk of bias, especially selection bias [60]. However, certain studies indicate that the results may be similarly robust [61].

## 5. Conclusions

In conclusion, the results of our meta-analysis suggest a significant relationship between HNC risk and short telomere length. This relationship may depend on the specific subtype of HNC. In our findings, the relationship was shown to be significant in overall HNCs and in oral cancers. Future research may focus on high-quality prospective studies of specific HNCs, which may validate our findings and delineate possible confounding factors. The present study is the first attempt at a systematic review and meta-analysis of existing data. Our results highlight the importance of further studies in this field to ascertain the role of telomere length measurement as a potential clinically significant biomarker for HNCs.

## Figures and Tables

**Figure 1 ijms-25-09000-f001:**
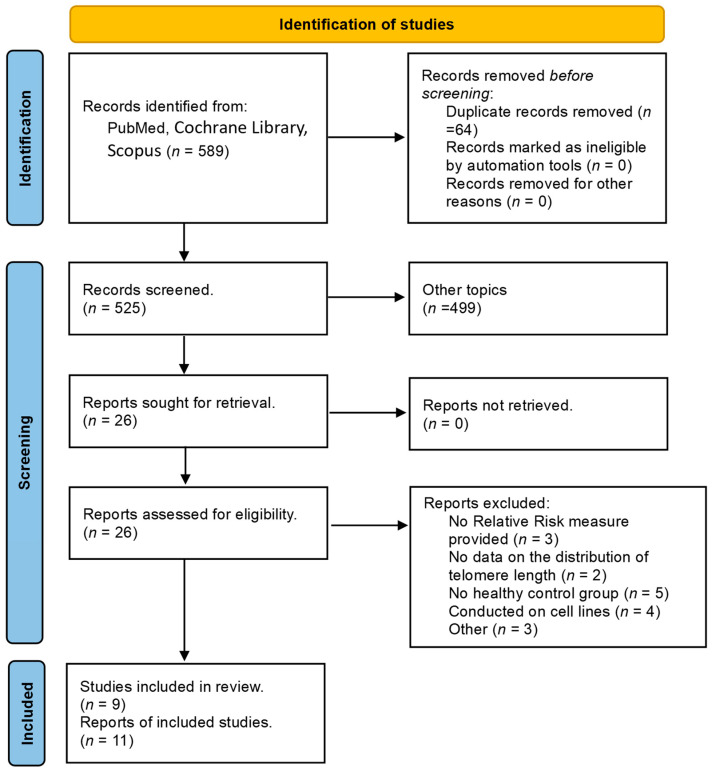
PRISMA flow diagram of the meta-analysis.

**Figure 2 ijms-25-09000-f002:**
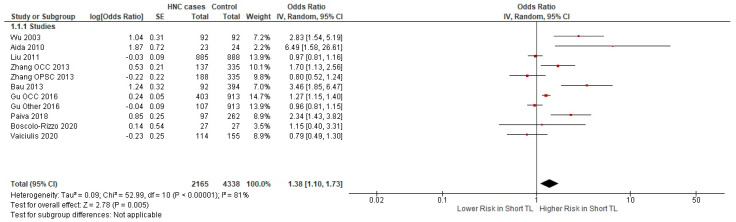
Forest plot representing the meta-analysis of the relationship between short TL and the risk for HNC using a random effects model [37,38,39,40,41,42,43,44,45]. Abbreviations: HNC = head and neck cancer; SE = standard error; CI = confidence interval; TL = telomere length. The black rhombus represents the combined OR, its width representing the 95% CI. The horizontal lines with the red square represent the 95% CI.

**Figure 3 ijms-25-09000-f003:**
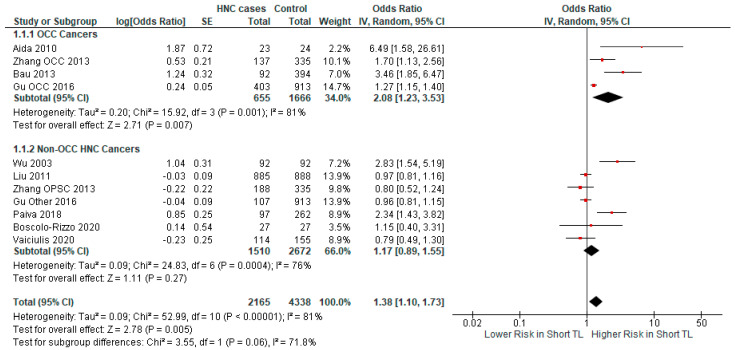
Forest plot representing the subgroup analysis of the relationship between short TL and the risk for HNC, categorized by HNC subtype [37,38,39,40,41,42,43,44,45]. Abbreviations: HNC = head and neck cancer; SE = standard error; CI = confidence interval; TL = telomere length; OCC = oral cavity cancer. The black rhombus represents the combined OR, its width representing the 95% CI. The horizontal lines with the red square represent the 95% CI.

**Figure 4 ijms-25-09000-f004:**
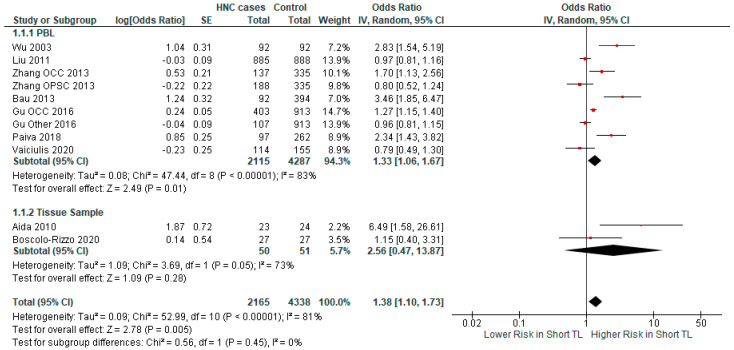
Forest plot representing the subgroup analysis of the relationship between short TL and the risk for HNC, categorized by source of DNA sample [37,38,39,40,41,42,43,44,45]. Abbreviations: HNC = head and neck cancer; SE = standard error; CI = confidence interval; TL = telomere length; PBL = peripheral blood leukocytes. The black rhombus represents the combined OR, its width representing the 95% CI. The horizontal lines with the red square represent the 95% CI.

**Figure 5 ijms-25-09000-f005:**
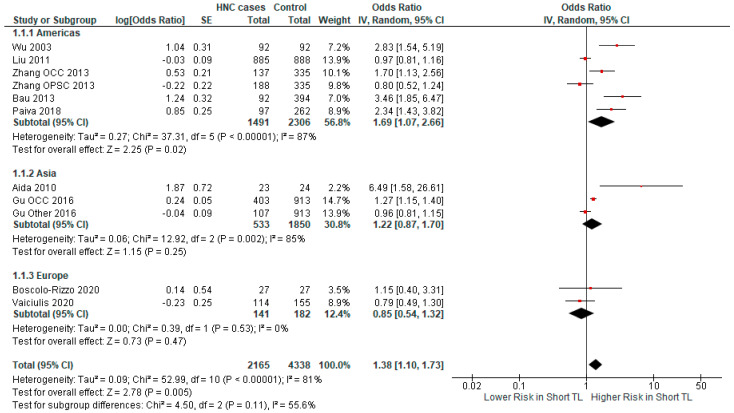
Forest plot representing the subgroup analysis of the relationship between short TL and the risk for HNC, categorized by study region [37,38,39,40,41,42,43,44,45]. Abbreviations: HNC = head and neck cancer; SE = standard error; CI = confidence interval; TL = telomere length. The black rhombus represents the combined OR, its width representing the 95% CI. The horizontal lines with the red square represent the 95% CI.

**Figure 6 ijms-25-09000-f006:**
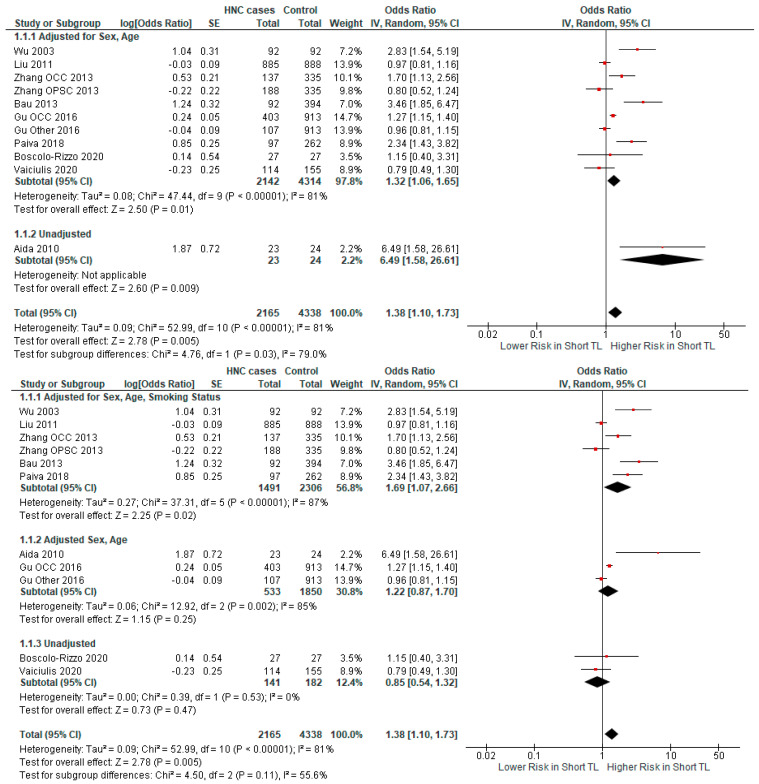
Forest plots representing the subgroup analysis of the relationship between short TL and the risk for HNC, categorized by factors the ORs have been adjusted for [37,38,39,40,41,42,43,44,45]. Abbreviations: HNC = head and neck cancer; SE = standard error; CI = confidence interval; TL = telomere length. The black rhombus represents the combined OR, its width representing the 95% CI. The horizontal lines with the red square represent the 95% CI.

**Figure 7 ijms-25-09000-f007:**
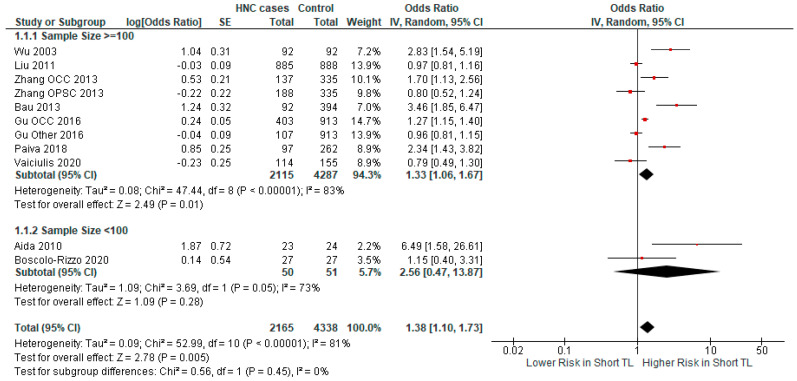
Forest plot representing the subgroup analysis of the relationship between short TL and the risk for HNC, categorized by sample size [37,38,39,40,41,42,43,44,45]. Abbreviations: HNC = head and neck cancer; SE = standard error; CI = confidence interval; TL = telomere length. The black rhombus represents the combined OR, its width representing the 95% CI. The horizontal lines with the red square represent the 95% CI.

**Figure 8 ijms-25-09000-f008:**
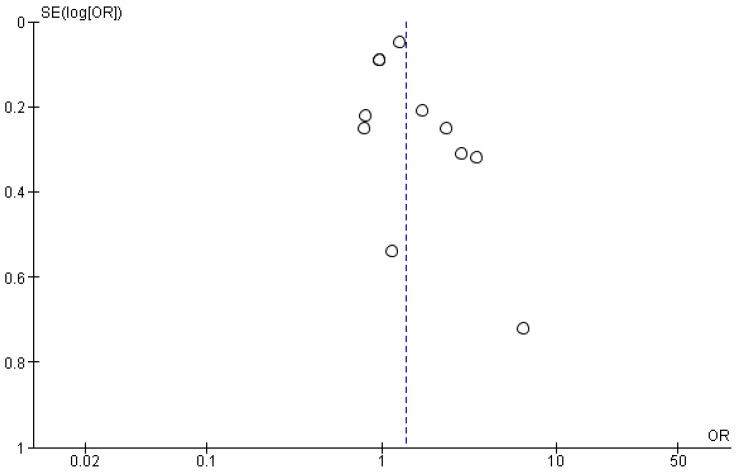
Funnel Plot of the studies included in the meta-analysis [37,38,39,40,41,42,43,44,45].

**Table 1 ijms-25-09000-t001:** Studies included in the meta-analysis. Characteristics of the included studies: first author and year the study was published, country the study was conducted in, types of cancer in the patient group, odds ratios and confidence intervals for the association between cancer and short TL, the sample size of the study, and the distribution of cases and controls.

Author	Year	Country	Cancer Type	Odds Ratio	Sample Size	Cases/ Controls
Wu [37]	2003	USA	Head and Neck	2.83	184	92/92
Aida [38]	2010	Japan	Oral—CIS	6.5	49	25/24
Liu [39]	2011	USA	Head and Neck	0.97	1773	885/888
Zhang [40]	2013	USA	OCC	1.7	523	137/335
Zhang [40]	2013	USA	OPC	0.8	472	188/335
Bau [41]	2013	USA	OSCC	3.47	486	92/394
Yayun Gu [42]	2016	China	Oral Cancer	1.28	1316	403/913
Yayun Gu [42]	2016	China	Other HNC sites	0.96	1020	107/913
Paiva [43]	2018	Brazil	Head and Neck	2.34	359	97/262
Boscolo-Rizzo [44]	2020	Italy	Head and Neck	1.15	27	27/27
Vaiciulis [45]	2020	Lithuania	Laryngeal	0.79	269	114/155

Abbreviations: CIS = carcinoma in situ; OCC = oral cavity cancer; OPC = oropharyngeal squamous cell carcinoma; OSCC = oral squamous cell carcinoma; TL = telomere length.

**Table 2 ijms-25-09000-t002:** Study design and methodological and demographic details of the included studies. Characteristics of the included studies: first author and year the study was published, the design the study followed, the source of the DNA used for the measurement of the TL, the methods used to measure the TL, the mean age and distribution of gender among the study subjects, and the factors the OR was adjusted for.

Author	Year	Study Design	DNA Source	Measurement Method	Mean Age	Gender (Percentage Male)	Adjusted
Wu [37]	2003	Retro	Leuko	Southern Blot	57.6 cases, 57.4 controls	96% cases, 96% controls	Age, sex, smoking
Aida [38]	2010	Retro	Tumor	Southern Blot	63.3 cases, 55.4 controls	75% cases, 74% controls	-
Liu [39]	2011	Retro	Leuko	Real-Time PCR	56.8 cases, 55.4 controls	74.8% cases, 74.4% controls	Age, sex, smoking, drinking
Zhang [40]	2013	Retro	Leuko	Real-Time PCR	-	74.2% cases ^1^, 80.3% controls	Age, sex, smoking, drinking
Zhang [40]	2013	Retro	Leuko	Real-Time PCR	-	74.2% cases ^1^, 80.3% controls	Age, sex, smoking, drinking
Bau [41]	2013	Retro	Leuko	Real-Time PCR	57.22 cases, 58.30 controls	59% cases, 57% controls	Age, sex, smoking, drinking
Yayun Gu [42]	2016	Retro	Leuko	Real-Time PCR	61.29 cases ^1^, 59.78 controls	70% cases ^1^, 74% controls	Age, sex, smoking, drinking
Yayun Gu [42]	2016	Retro	Leuko	Real-Time PCR	61.29 cases ^1^, 59.78 controls	70% cases ^1^, 74% controls	Age, sex, smoking, drinking
Paiva [43]	2018	Retro-Cohort	Leuko	Rreal-Time PCR	55 cases	89% cases	Age, sex
Boscolo-Rizzo [44]	2020	Retro	Tumor	Real-Time PCR	66 cases	73% cases	Age, sex, smoking, drinking
Vaiciulis [45]	2020	Retro	Leuko	Real-Time PCR	62.8 cases, 62.5 controls	96.3% cases, 96.7% controls	Age, sex

Abbreviations: TL = telomere length; OR = odds ratio; Retro = retrospective; Leuko = peripheral blood leukocytes; PCR = polymerase chain reaction. ^1^ The overall mean age and gender distribution for all the cases is presented.

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
