# Peer review of "The Association between Telomere Length and Head and Neck Cancer Risk: A Systematic Review and Meta-Analysis"

_ijms, 2024, doi:10.3390/ijms25169000_

Round 1
Reviewer 1 Report
Comments and Suggestions for Authors
This is a thorough and thoughtful review on the relationship between telomere length and clinical variables.
- Please clarify that the definitions of short and long telomere lengths were consistent across the studies, that even with the same technique (eg. real-time PCR), the cutoffs were of acceptable consistency.
- For the studies that examined the tumor tissues, did the authors indicate whether tumor collection was at the time of diagnosis or after treatment? Would this confound the analysis?
- What about staging? Is there an association between telomere length and staging?
Author Response
Dear Reviewer,
We are grateful for your valuable comments and suggestions, we strongly believe they improved the quality of our work. We have carefully considered each of your suggestions and have made the necessary revisions accordingly.
In addition to addressing the specific points raised, we have also written and included an abstract based on the PRISMA checklist, supplementing the original abstract.
All changes are clearly tracked using the embedded in MS Word tool.
Below, we provide a detailed account of the changes made in response to your comments
Comment:
Please clarify that the definitions of short and long telomere lengths were consistent across the studies, that even with the same technique (e.g., real-time PCR), the cutoffs were of acceptable consistency.
Response:
We thank you for raising this point. In our meta-analysis the definitions of short and long telomere lengths were consistently based on the median telomere length. All the included studies followed the methodological standards of similar peer-reviewed meta-analysis in the field (eg doi: 10.1371/journal.pone.0133174), utilizing the median telomere length as the cutoff point. Additionally, we further restricted our methodology by using the median telomere length of the controls as the cutoff point. Our stricter criterion was achieved in all our studies, except Zhang et al. Zhang et al provided the results using the median of all subjects as the cutoff point and did not provide the necessary data for the derivation of the results using the median TL of controls. Zhang et al. is a pivotal study in the field, and its inclusion follows the standards of similar peer-reviewed meta-analysis. Finally, we modified the discussion segment to clearly mention and discuss this limitation.
Comment:
For the studies that examined the tumor tissues, did the authors indicate whether tumor collection was at the time of diagnosis or after treatment? Would this confound the analysis?
Response:
We thank the reviewer for this important question. It can be strongly inferred from the text that the collection preceded any treatment, even though not all authors included an explicit statement regarding the time of collection as it relates to treatment.
Comment:
What about staging? Is there an association between telomere length and staging?
Response:
We thank the reviewer for this interesting comment. Unfortunately, there is not enough data to perform a subgroup analysis on the basis of stage.

Reviewer 2 Report
Comments and Suggestions for Authors
This study conducted a meta-analysis for the association of telomere length and risk of head and neck cancers. It also performed several sub-group analyses. The telomere length was classified into long and short groups as exposure. A major concern is that participating studies used the median value, instead of absolute cutoff, to define the exposure. This review study did not provide a summary of the distribution/median value of the participating studies, and it did not discuss this as a limitation. All comments are as below:
Abstract, line 19: Was long TL the reference group for the reported ORs? If so, the definition of long and short TL should be mentioned in the abstract.
Abstract, line 20: The observed heterogeneity should be mentioned in the abstract.
Introduction, line 59: It’s better to mention the cancer types in the main text, so that the readers do not have to check the references for these cancer types.
Introduction, line 64: It’s better to use “pathogenic mutations at the protein-coding genes” instead of “mutation of proteins”.
Methods, line 106: What was the definition of long and short TL for this review? Did all studies included in the analyses use the same definition?
Results, line 170: “The median TL was considered as the cut-off point for defining the short TL 170 group when calculating the OR.” The studies in the meta-analyses used their own median TL, instead of an absolute value as cutoff. Did their median value vary a lot across the studies? It will be better if the range/median/SD for these studies can be summarized. This may partially explain the observed heterogeneity, and this should be included in the discussion for the limitations.
Results, line 224: Meta-analyses were conducted stratified by sample size of the participating studies (>=100 or <100). In the figure 3, the Boscolo-Rizzo 2020 study had 114 cases and 155 controls. Why was this study classified as sample size <100? Similarly, the Vaiciulis 2020 study had 27 cases and 27 controls, but it was classified as sample size >=100.
Discussion, line 241-250: This paragraph seems confusing. The first paragraph of the discussion usually summarizes the study. However, this paragraph discussed the associations of TL with other cancers, which was not the focus of this study. These contents may move into the introduction.
Author Response
Dear Reviewer,
We are grateful for your valuable comments and suggestions, we strongly believe they improved the quality of our work. We have carefully considered each of your suggestions and have made the necessary revisions accordingly.
In addition to addressing the specific points raised, we have also written and included an abstract based on the PRISMA checklist, supplementing the original abstract.
All corrections/modifications are clearly tracked using the embedded in MS Word tool. We included a fully tracked and a clean copy of manuscript. The line numbers in our replies reflect the fully tracked version.
Below, we provide a detailed account of the changes made in response to your comments
General Comment:
A major concern is that participating studies used the median value, instead of absolute cutoff, to define the exposure. This review study did not provide a summary of the distribution/median value of the participating studies, and it did not discuss this as a limitation
Response:
Thank you for raising this valuable point. Detailed explanations to these questions are included in replies to the following comments: “Methods, line 106: What was the definition of long and short TL for this review? Did all studies included in the analyses use the same definition?”, “Results, line 170: “The median TL was considered as the cut-off point for defining the short TL 170 group when calculating the OR.” The studies in the meta-analyses used their own median TL, instead of an absolute value as cutoff. Did their median value vary a lot across the studies? It will be better if the range/median/SD for these studies can be summarized. This may partially explain the observed heterogeneity, and this should be included in the discussion for the limitations.”
Comment:
Abstract, line 19: Was long TL the reference group for the reported ORs? If so, the definition of long and short TL should be mentioned in the abstract.
Response:
We thank the reviewer for pointing this out. We agree with your comment and we have modified the manuscript accordingly, in line 19.
Comment:
Abstract, line 20: The observed heterogeneity should be mentioned in the abstract.
Response:
We thank the reviewer for this suggestion. We have mentioned the observed heterogeneity in the abstract, at line 21.
Comment:
Introduction, line 59: It’s better to mention the cancer types in the main text, so that the readers do not have to check the references for these cancer types.
Response:
We thank the reviewer for this suggestion. We have modified the manuscript to include the cancer types in the main text, in lines 68-74.
Comment:
Introduction, line 64: It’s better to use “pathogenic mutations at the protein-coding genes” instead of “mutation of proteins”.
Response:
We thank the reviewer for this suggestion. We have updated the text to "pathogenic mutations at the protein-coding genes", see line 84.
Comment:
Methods, line 106: What was the definition of long and short TL for this review? Did all studies included in the analyses use the same definition?
Response:
We thank the reviewer for raising this point. In our meta-analysis the definitions of short and long telomere lengths were consistently based on the median telomere length. All the included studies followed the methodological standards of similar peer-reviewed meta-analysis in the field (eg doi: 10.1371/journal.pone.0133174), utilizing the median telomere length as the cutoff point. Additionally, we further restricted our methodology by using the median telomere length of the controls as the cutoff point. Our stricter criterion was achieved in all our studies, except Zhang et al. Zhang et al provided the results using the median of all subjects as the cutoff point and did not provide the necessary data for the derivation of the results using the median TL of controls. Zhang et al. is a pivotal study in the field, and its inclusion follows the standards of similar peer-reviewed meta-analysis. Finally, we modified the discussion segment to clearly mention and discuss this limitation.
Comment:
Results, line 170: “The median TL was considered as the cut-off point for defining the short TL 170 group when calculating the OR.” The studies in the meta-analyses used their own median TL, instead of an absolute value as cutoff. Did their median value vary a lot across the studies? It will be better if the range/median/SD for these studies can be summarized. This may partially explain the observed heterogeneity, and this should be included in the discussion for the limitations.
Response:
We thank the reviewer for this suggestion. We concur that such an addition could have provided useful information. However, the studies included in our meta-analysis had widely heterogeneous approaches to reporting TL measurements (RTL – relative telomere length, age-adjusted relative LTL – leukocyte telomere length, normalized TCRa (NTCR), T/S median, etc.) and some did not include any direct measurements of TL. Therefore, no meaningful analysis could be conducted, and this information has not been included. We have also stated this as a possible limitation of our analysis. Line 402-406.
Comment:
Results, line 224: Meta-analyses were conducted stratified by sample size of the participating studies (>=100 or <100). In the figure 3, the Boscolo-Rizzo 2020 study had 114 cases and 155 controls. Why was this study classified as sample size <100? Similarly, the Vaiciulis 2020 study had 27 cases and 27 controls, but it was classified as sample size >=100.
Response:
We thank the reviewer for pointing this typo. We have corrected the numerical mistake and modified the manuscript accordingly. (page 10, figure 3f. and line 373)
Comment:
Discussion, line 241-250: This paragraph seems confusing. The first paragraph of the discussion usually summarizes the study. However, this paragraph discussed the associations of TL with other cancers, which was not the focus of this study. These contents may move into the introduction.
Response:
We thank the reviewer for pointing this comment. We agree with the recommendation and therefore modified the manuscript accordingly. (see lines 290-298 and lines 68-76)

Reviewer 3 Report
Comments and Suggestions for Authors
Authors did a metaanalyses as it should be done.
They clearly expalined how literature included was selected.
They clearly provide how they evaluated the results.
The data is discussed in all possible directions and is selfcritical.
shortcuts of the study are clearly highlighted in the discussion.
conclusions are carefully made and it is stated that the found correlation between telomere lenght and head ans neck cancer is there , but still the biological meaningis not understood yet.
Even the studies included still show (as authors state themselves) some heterogeneity, the point is that here for the first time shortened telomerelenght could be also found in head and neck cancer - as in other cancers before. Also authors showed that adjustments for age, sex, and smoking did not affect the significance of their findings.
Overall, an important study in the field.
Author Response
We would like to thank the reviewer for the supportive comments and remarks on our work.

Reviewer 4 Report
Comments and Suggestions for Authors
In the submitted manuscript authors conducted a meta-analysis to assess the association of telomere length (TL) with a risk for head and neck cancers (HNC), and found a significant relationship between short TL and increased HNC risk (OR 1.38, 95% CI: 1.10-1.73, p=0.005), or more specifically, risk for oral cancer, also showing that age, sex, and smoking do not affect that association.
This manuscript has presented interesting results, and used methodology is proper and robust; however, there are many drawbacks which must be corrected or further improved.
1) In 'Introduction', more information should be provided on what all affect TL length, both in physiological conditions and especially cancer (e.g., DOI: 10.1038/s42003-022-03521-7). Also results should be discussed in that light, i.e., is there something special related to HNC and especially oral cancer etiopathology that could cause short TL?! Furthermore, methods used for measuring TL should be briefly presented.
2) Prisma statement must be properly referenced (DOI: 10.1136/bmj.n71), while valid URLs must be provided for all used on-line databases.
3) Lines 106-107: Using median TL as a cut-off value should be mentioned already in this sentence.
4) Text in those blue rectangles in Figure 1 would be more readable if it is written vertically.
5) There is no point to present ORs and 95% CIs in Table 1 (and even use 3 columns!) since all these are presented also in Figure 2 (where it should be). Instead, it would be more interesting to provide median TLs for both cases and controls.
Also, in Table 2, those two studies which have double results (Zhang 2013 and Yayun Gu 2016) cannot have the same mean age and percentage of males for each sub-type of HNC cases since, as it is written in Table 1, numbers of cases were different!
Furthermore, it is unclear why total sample size for Zhang 2013 differs (523 vs. 472) and for Yayun Gu 2016 doesn't (1423 for both)?!
6) Since Table 2 are showing different things than Table 1, it is not "cont." and each table should have proper caption. Furthermore, order of studies should be the same in tables and forest plots!
7) Text in figures is blurry are generally hard to read, while figures should have higher resolution.
8) In Figure 2, there is no point to present subtotal and total results since they are completely the same, i.e., there are no subtotal results without subgroup analysis!
9) Figures 3 should be properly ordered and named, i.e., multi-panel figures should be drawn as such and sub-panels should be named as Fig.3a, 3b, 3c, etc., and sub-panels should not be divided by the main text.
10) If results mentioned in lines 235-237 were not shown, this must be stated as "(data not show)" at the end of that sentence.
11) Lines 279-282: Presenting statistically unsignificant results as something meaningful is meaningless!
12) Lines 282-284: You really haven't proven in any possible way that "tumor TL is associated with leukocyte TL"!!!
13) Lines 320-321: Statement that "In our meta-analysis all studies had a sample size equal or greater than 50" is simply false since study by Aida 2010 had 49 samples, while in Table 1 it was written that Bosco-lo-Rizzo
2020 had even less, 27, but I presume there should be 54!
14) References 17, 21, 29, and 40 should be written like all others.
Comments on the Quality of English Language1) Lines 56-57: Statement "while genomic instability due to the shorter telomeres induces to carcinogenic mutations" is unclear, so I presume "to" is redundant.
2) Words correlation/correlated should be replaced with association/associated since I presume they were not used in the context of calculated correlation coefficient.
3) Line 68: "predictive biomarkers" are those that affect efficacy of cancer treatment, so TL in your case could be considered only as a "risk biomarker"
4) Line 81: Proper would be "the full text of any paper".
5) All phrases build with common nouns like odds ratio, forest plot, funnel plot, etc. should not be written with the first capital letter(s) when are in the middle of a sentence.
6) Line 95: The complete phrase "English language" would suite better.
7) Lines 101-102: Statement "If an article presented data from multiple studies, we considered them as independent studies." is confusing since Zhang 2013 and Yayun Gu 2016 just analyzed two different types of HNC, they are not separate studies!
8) Line 126: It is unclear what is "bilateral" P value, but proper is "two-tail"!
9) Line 160: Proper is "its subjects".
10) Authors should re-check text that all abbreviations were explained in both 'Abstract' and main text (e.g., PBL).
11) All abbreviations presented in both figures and tables must be explained in figure legends and table footnotes, respectively.
Author Response
Dear Reviewer,
We are grateful for your valuable comments and suggestions, we strongly believe they improved the quality of our work. We have carefully considered each of your suggestions and have made the necessary revisions accordingly.
In addition to addressing the specific points raised, we have also written and included an abstract based on the PRISMA checklist, supplementing the original abstract.
All corrections/modifications are clearly tracked using the embedded in MS Word tool. We included a fully tracked and a clean copy of manuscript. The line numbers in our replies reflect the fully tracked version.
Below, we provide a detailed account of the changes made in response to your comments.
Comment:
In 'Introduction', more information should be provided on what all affect TL length, both in physiological conditions and especially cancer (e.g., DOI: 10.1038/s42003-022-03521-7). Also, results should be discussed in that light, i.e., is there something special related to HNC and especially oral cancer etiopathology that could cause short TL? Furthermore, methods used for measuring TL should be briefly presented.
We thank you for this suggestion. We have expanded the Introduction to include additional information on factors affecting TL length in both physiological conditions and cancer. The changes can be found in lines 54-62.
We thank the reviewer for giving us the opportunity to clarify this issue. As oral cavity is an anatomical part of the head and neck (https://www.cancer.gov/types/head-and-neck/head-neck-fact-sheet), it is affected similarly by same risk factors and cancer occurrence like other areas within (eg pharynx and larynx). Particularly, the most frequent type of head-neck cancer is head and neck squamous cell carcinoma (HNSCC) affecting all its anatomical parts (Boscolo-Rizo et al, Cancer Metastasis Rev 2016). To the best of our knowledge the mechanisms leading to telomere attrition in HNSCC appear to be common within all these anatomic parts of head and neck (Boscolo-Rizo et al, Cancer Metastasis Rev 2016).
We thank for the suggestion. We have now briefly described the methods used for measuring TL in lines 52-54.
Comment:
Prisma statement must be properly referenced (DOI: 10.1136/bmj.n71), while valid URLs must be provided for all used on-line databases.
Response:
Thank you for highlighting this. We have added the proper reference for the Prisma statement and provided valid URLs for all online databases used. This change can be found in line 95.
Comment:
Lines 106-107: Using median TL as a cut-off value should be mentioned already in this sentence.
Response:
We thank the reviewer for the suggestion. We have revised the text to mention the use of median TL as a cut-off point in line 129.
Comment:
Text in those blue rectangles in Figure 1 would be more readable if it is written vertically.
Response:
We appreciate your remark. We have modified the text in Figure 1 to be vertical for better readability.
Comment:
There is no point to present ORs and 95% CIs in Table 1 (and even use 3 columns!) since all these are presented also in Figure 2 (where it should be). Instead, it would be more interesting to provide median TLs for both cases and controls.
Response:
We thank the reviewer for this suggestion. We concur that such an addition could have provided useful information. However, the studies included in our meta-analysis had widely heterogeneous approaches to reporting TL measurements (RTL – relative telomere length, age-adjusted relative LTL – leukocyte telomere length, normalized TCRa (NTCR), T/S median, etc.) and some did not include any direct measurements of TL. Therefore, no meaningful analysis could be conducted, and this information has not been included. We have also stated this as a possible limitation of our analysis. Line 402-406.
Comment:
Also, in Table 2, those two studies which have double results (Zhang 2013 and Yayun Gu 2016) cannot have the same mean age and percentage of males for each sub-type of HNC cases since, as it is written in Table 1, numbers of cases were different!
Response:
Thank you for pointing this out, we agree with your comment and we have modified the manuscript accordingly by adding a footnote explaining that these numbers are the overall mean age and gender distribution for all the cases is presented. Unfortunately, more detailed demographic information was not included in these studies.
Comment:
Furthermore, it is unclear why total sample size for Zhang 2013 differs (523 vs. 472) and for Yayun Gu 2016 doesn't (1423 for both)?
Response:
We thank the reviewer for this remark. We have corrected the sample size discrepancies for Zhang 2013 in Table 1.
Comment:
Since Table 2 are showing different things than Table 1, it is not "cont." and each table should have proper caption. Furthermore, order of studies should be the same in tables and forest plots!
Response:
We appreciate the reviewer’s feedback. We have revised the captions for Table 2 and ensured that the order of studies is consistent across tables and forest plots.
Comment:
Text in figures is blurry are generally hard to read, while figures should have higher resolution.
Response:
Thank you for bringing this to our attention. We have re-run our software to produce higher resolution figures for better readability.
Comment:
In Figure 2, there is no point to present subtotal and total results since they are completely the same, i.e., there are no subtotal results without subgroup analysis!
Response:
We appreciate your suggestion. We have revised Figure 2 to remove the redundant subtotal results.
Comment:
Figures 3 should be properly ordered and named, i.e., multi-panel figures should be drawn as such and sub-panels should be named as Fig.3a, 3b, 3c, etc., and sub-panels should not be divided by the main text.
Response:
We thank the reviewer for the constructive remark. We have reorganized the text and properly named the multi-panel figures as suggested.
Comment:
If results mentioned in lines 235-237 were not shown, this must be stated as "(data not shown)" at the end of that sentence.
Response:
We thank the reviewer for the suggestion. We have added "(data not shown)" at the end of the sentence, line 282.
Comment:
Lines 279-282: Presenting statistically insignificant results as something meaningful is meaningless!
Response:
We thank the reviewer for the critical observation. We have revised the text to clarify the interpretation, in lines 327-328.
Comment:
Lines 282-284: You really haven't proven in any possible way that "tumor TL is associated with leukocyte TL"!
Response:
We thank the reviewer for this critical comment. We have rewritten the paragraph in the discussion to more accurately reflect our findings and observations, in lines 330-339.
Comment:
Lines 320-321: Statement that "In our meta-analysis all studies had a sample size equal or greater than 50" is simply false since study by Aida 2010 had 49 samples, while in Table 1 it was written that Bosco-lo-Rizzo 2020 had even less, 27, but I presume there should be 54!
Response:
Thank you for highlighting this discrepancy. We have corrected the text, in line 372.
Comment:
References 17, 21, 29, and 40 should be written like all others.
Response:
Thank you for the comment. We updated these references to match the format of the others.
Comments on the Quality of English Language
Comment:
Lines 56-57: Statement "while genomic instability due to the shorter telomeres induces to carcinogenic mutations" is unclear, so I presume "to" is redundant.
Response:
Thank you for your suggestion. We have removed the redundant "to", line 67.
Comment:
Words correlation/correlated should be replaced with association/associated since I presume they were not used in the context of calculated correlation coefficient.
Response:
Thank you for pointing this out. We have replaced "correlation/correlated" with "association/associated" throughout the manuscript.
Comment:
Line 68: "predictive biomarkers" are those that affect efficacy of cancer treatment, so TL in your case could be considered only as a "risk biomarker."
Response:
We thank the reviewer for point this correction. We have updated the terminology to "risk biomarker", in line 89.
Comment:
Line 81: Proper would be "the full text of any paper."
Response:
We thank the reviewer for pointing this suggestion. We have revised the sentence to "the full text of any paper", in line 162.
Comment:
All phrases build with common nouns like odds ratio, forest plot, funnel plot, etc. should not be written with the first capital letter(s) when are in the middle of a sentence.
Response:
We thank the reviewer for this observation. We have ensured that common nouns like "odds ratio," "forest plot," and "funnel plot" are not capitalized when used in the middle of a sentence.
Comment:
Line 95: The complete phrase "English language" would suit better.
Response:
We thank the reviewer for the suggestion. We have replaced "English" with "English language", in line 118.
Comment:
Lines 101-102: Statement "If an article presented data from multiple studies, we considered them as independent studies." is confusing since Zhang 2013 and Yayun Gu 2016 just analyzed two different types of HNC, they are not separate studies!
Response:
Thank you for this observation. We have clarified this statement, in line 125.
Comment:
Line 126: It is unclear what is "bilateral" P value, but proper is "two-tail"!
Response:
We thank the reviewer for the suggestion. We have replaced "bilateral" with "two-tail", in line 152.
Comment:
Line 160: Proper is "its subjects".
Response:
We have corrected the phrase to "its subjects", in line 192.
Comment:
Authors should re-check text that all abbreviations were explained in both 'Abstract' and main text (e.g., PBL).
Response:
We thank the reviewer for the suggestion. We have ensured that all abbreviations are explained in both the Abstract and the main text.
Comment:
All abbreviations presented in both figures and tables must be explained in figure legends and table footnotes, respectively.
Response:
Thank you for the suggestion. We have added explanations for all abbreviations in figure legends and table footnotes.

Reviewer 5 Report
Comments and Suggestions for Authors
Specific comments to the authors
The submitted review "The association between telomere length and head and neck cancer risk: a systematic review and meta-analysis" collects, summarises and analyses the relationship between telomere length and the risk of head and neck cancer (HNC) based on published studies.
The topics presented range from classic clinic-pathological findings/characteristics and treatment concepts of HNC to telomeres and telomere length in relation to cancer now and in the future. The consecutive meta-analysis shows a significant association between HNC risk and TL in general as well as for subgroups analysing oral cancer, DNA sample source, study region, risk factor adjustment and sample size. In conclusion, the author provides an interesting meta-analysis for telomeres and telomere length in relation to HCN, which is mostly easy to read, follow and understand. The authors should clarify some aspects before accepting the manuscript for publication, as mentioned below.
# Title: As the submitted manuscript is a meta-analysis rather than a review, the term "systematic review" should be deleted.
# Figure 1: Please add the applied PRISMA criteria in a supplement. The number of included studies of n=9 and n=11 is not (?) congruent with the number of studies in Table 1. Please check.
# Figure 2: Please add the character of the studies mentioned (retrospective/prospective, clinical trials (phase)).
# Figure 3.1: Please specify the term "HCN other".
# Figure 3.4: Please include TNM/UICC stage, p16, and HPV status as definitive risk factors for HCN in the meta-analysis.
Comments on the Quality of English LanguageMinor editing of English language required.
Author Response
Dear Reviewer,
We are grateful for your valuable comments and suggestions, we strongly believe they improved the quality of our work. We have carefully considered each of your suggestions and have made the necessary revisions accordingly.
In addition to addressing the specific points raised, we have also written and included an abstract based on the PRISMA checklist, supplementing the original abstract.
All corrections/modifications are clearly tracked using the embedded in MS Word tool. We included a fully tracked and a clean copy of manuscript. The line numbers in our replies reflect the fully tracked version.
Below, we provide a detailed account of the changes made in response to your comments
Comment:
Title: As the submitted manuscript is a meta-analysis rather than a review, the term "systematic review" should be deleted.
Response:
We thank the reviewer for pointing this out. Nevertheless, the title identifies the study as a systematic review per the prisma checklist.
Comment:
Figure 1: Please add the applied PRISMA criteria in a supplement.
Response:
Thank you for your suggestion. We have added the PRISMA checklist as a supplement.
Comment:
The number of included studies of n=9 and n=11 is not congruent with the number of studies in Table 1. Please check.
Response:
We thank the reviewer for pointing this out. We agree with your comment, 9 articles are analyzed that include 11 reports/datasets. We have modified the manuscript to clarify that, in line 164.
Comment:
Figure 2: Please add the character of the studies mentioned (retrospective/prospective, clinical trials (phase)).
Response:
We thank the reviewer for the suggestion. The study design is included in Table 2.
Comment:
Figure 3.1: Please specify the term "HCN other".
Response:
We thank the reviewer for the suggestion. We agree with the comment and therefore modified the figure accordingly.
Comment:
Figure 3.4: Please include TNM/UICC stage, p16, and HPV status as definitive risk factors for HCN in the meta-analysis.
Response:
We thank the reviewer for the suggestion. Unfortunately, the studies did not include enough data on TNM/UICC stage, p16, or HPV status for a meaningful analysis to be performed. We appreciate your excellent suggestion and believe future studies should include such critical data. We have also added these issues to the limitations section. Lines 399-401.

Round 2
Reviewer 2 Report
Comments and Suggestions for Authors
All my concerns have been resolved.
Reviewer 4 Report
Comments and Suggestions for Authors
Authors have satisfactorily answered to all my concerns and substantially improved quality of this manuscript through revision.